# Molecular Regulation of Yak Preadipocyte Differentiation and Proliferation by *LncFAM200B* and ceRNA Regulatory Network Analysis

**DOI:** 10.3390/cells11152366

**Published:** 2022-08-01

**Authors:** Hongbiao Ran, Youzhualamu Yang, Mengning Luo, Xinrui Liu, Binglin Yue, Zhixin Chai, Jincheng Zhong, Hui Wang

**Affiliations:** Key Laboratory of Qinghai-Tibetan Plateau Animal Genetic Resource Reservation and Utilization, Sichuan Province and Ministry of Education, Southwest Minzu University, Chengdu 610225, China; 200710002007@stu.swun.edu.cn (H.R.); 210710002002@stu.swun.edu.cn (Y.Y.); 210710002005@stu.swun.edu.cn (M.L.); 200710002009@stu.swun.edu.cn (X.L.); 80300193@swun.edu.cn (B.Y.); 22100108@swun.edu.cn (Z.C.)

**Keywords:** preadipocytes, differentiation, proliferation, *lncFAM200B*, ceRNA, Yak

## Abstract

The positive regulatory role of *lncFAM200B* in differentiation and lipid deposition in yak intramuscular preadipocytes has been demonstrated in our previous study. However, the regulatory mechanisms remain unclear. In this study, we aimed to produce complete mRNA and microRNA (miRNA) profiles after adenovirus-mediated *lncFAM200B* overexpression in yak preadipocytes using high-throughput sequencing. We constructed a competing endogenous RNA (ceRNA) network with *lncFAM200B* as the core and identified the functions of the selected target miRNA during cell proliferation and differentiation. We obtained 118 differentially expressed genes (DEGs) after *lncFAM200B* overexpression, 76 of which were up-regulated, including Notch signaling members *NOTCH3*, *DTX3L*, and *HES4*, and 42 DEGs were down-regulated, including genes related to the cell cycle (*CCNA2*, *BUB1*, *CDC20*, *TOP2A*, and *KIF20A*). Additionally, many ubiquitin-mediated proteolysis pathway members were also significantly up-regulated (*BUA7*, *PML*, *TRIM21*, and *TRIM25*). MiRNA sequencing showed that 13 miRNAs were significantly up-regulated, and 12 miRNAs were down-regulated. Among them, 29 targets of 10 differentially expressed miRNAs (DEMs) were differentially expressed, including miR-152-*FBXO33*, miR-6529a-*TRIM21*, miR-148c-*NOTCH3*, and the miR-6529b-*HES4* axis. We further verified that overexpression and inhibition of miR-6529a can inhibit and promote, respectively, the proliferation and differentiation of preadipocytes. Taken together, our study not only revealed the regulatory network of *lncFAM200B* during yak preadipocytes differentiation but also laid a foundation for elucidating the cause for lower intramuscular fat content in yaks at the molecular level.

## 1. Introduction

Yaks (*Bos grunniens*) occur around the Qinghai-Tibet Plateau in China, and they are a primary source of income for herders and an important source of beef. Yaks are one of the oldest cattle breeds in high-altitude areas with a low domestication level [1], and they have a lower fat content than cattle. Fat content plays a crucial role in terms of beef flavor; hence, it has become a focus of consumers [2,3]. In recent years, breeding high-quality yak varieties to improve meat quality and production capacity has become a focal point of research.

Adipose tissue, the main form of energy storage in the body, plays an important role in the survival, reproduction, and lactation of yaks. Animal adipogenesis is a complex process that is precisely regulated by multiple factors [4,5,6], and it is affected by genetic and environmental factors. Intramuscular fat (IMF) content, the best indicator of beef grade [2], is a key aspect for improving meat quality [5]. The proliferation and differentiation levels of preadipocytes determine their biological processes and functions. For this reason, intramuscular preadipocytes have become the main focus of animal adipogenesis-related research in vitro. Numerous of studies have shown that both microRNAs (miRNAs) and long non-coding RNAs (lncRNAs) play precise regulatory roles in adipocyte proliferation and differentiation [4,7]. LncRNAs can regulate various biological processes through epigenetic, transcriptional, and post-transcriptional mechanisms [8,9]. MiRNAs primarily bind to the 3′-untranslated region sequence of an mRNA, leading to translation inhibition or gene splicing [7]. Moreover, a competing endogenous RNA (ceRNA) regulatory network exists among mRNAs, miRNAs, and lncRNAs [10]. As molecular sponges, lncRNAs can bind to miRNAs by competing with mRNA and other RNAs [3,10,11].

*LncFAM200B* is an lncRNA with a length of 472 nucleotides (nt), which was first identified in Qinchuan cattle through high-throughput RNA sequencing (RNA-Seq) [11]. *LncFAM200B* expression is markedly higher in fat tissue than in muscle tissue, and it inhibits proliferation of preadipocytes [4]. We previously cloned yak *lncFAM200B* and analyzed its function in preadipocytes during differentiation, showing that *lncFAM200B* is an lncRNA with a length of 531 nt in yaks, and its overexpression significantly improved the differentiation of intramuscular preadipocytes and promoted lipid deposition in yaks; moreover, it significantly inhibited cell proliferation. Suppressing the expression of *lncFAM200B* has the opposite effect [12,13]. However, the regulatory mechanisms underlying *lncFAM200B* overexpression effects on intramuscular preadipocyte differentiation and its targets remain unknown.

In this study, intramuscular preadipocytes were isolated from yak *longissimus dorsi* muscle tissue, and RNA-Seq was used to analyze differentially expressed gene (DEG) and differentially expressed miRNA (DEM) profiles due to *lncFAM200B* overexpression. The ceRNA regulatory network was constructed with *lncFAM200B* as the core by integrative analyses, and a target miRNA was selected for functional validation. We aimed to reveal the regulatory mechanism of *lncFAM200B* on lipid deposition or cell growth during preadipocyte differentiation, and our results lay the foundation for elucidating the cause of lower IMF content in yaks at the molecular level.

## 2. Materials and Methods

### 2.1. Adenovirus Generation and Cell Culture

An overexpression adenovirus vector, with a full-length yak *lncFAM200B* sequence, was constructed for *lncFAM200B* overexpression in vitro, here referred to as Ad_lnc200B. Briefly, the *lncFAM200B* sequence was cloned from the yak genome and was inserted into the pAdEasy-EF1-MCS-CMV-EGFP expression vector through double-enzyme digestion (*Kpn* I and *Xho* I). The high-purity vector was transfected into 293A cells using *Lipofiter*^TM^ reagent (HANBIO, Shanghai, China) after linearization with *Pac* I. The adenovirus was collected and used for infection after quality control, and empty-vector adenovirus was used as a negative control (NC, here referred to as Ad_G). Yak preadipocytes were isolated from *longissimus*
*dorsi* muscle tissue, according to a previous study [5], and were cultured at 37 °C and 5% CO_2_ in an incubator under sterile conditions. When the preadipocytes reached approximately 80% confluence, Ad_lnc200B and Ad_G were injected into the preadipocytes, and each treatment was performed using three biological replicates. Infected cells were cultured in differentiation-inducing medium (complete medium containing 50 µM oleic acid) for 6 h after infection, and the cells were collected for determination two days after.

### 2.2. Total RNA Extraction, Sequencing, and Data Processing

Total RNA was extracted using TRIzol reagent (Ambion/Invitrogen, Carlsbad, CA, USA) according to the manufacturer’s instructions, and the total amounts and integrity of RNA were assessed using an RNA Nano 6000 Assay Kit with the Bioanalyzer 2100 system (Agilent Technologies, Santa Clara, CA, USA). After quality examination, total RNA was used for library preparation. Briefly, the mRNA was purified from total RNA using poly-T oligo-attached magnetic beads. First strand cDNA was synthesized using random hexamer primers and M-MuLV Reverse Transcriptase (New England Biolabs, Ipswitch, MA, USA), and second-strand cDNA synthesis was subsequently performed using DNA polymerase I and dNTPs. After adenylating the 3′-ends of DNA fragments and ligating an adaptor with a hairpin loop structure, the library fragments were purified using the AMPure XP system (Beckman Coulter, Brea, CA, USA). After PCR amplification, the PCR product was purified using AMPure XP beads, resulting in the final library. The small RNA libraries were prepared from total RNA using NEBNext^®^ Multiplex Small RNA library Prep Set for Illumina^®^ (New England Biolabs) according to the manufacturer’s instructions.

Libraries were quantified using Qubit 2.0 Fluorometer (Thermo Fisher Scientific, Waltham, MA, USA), and insert size was determined using Agilent 2100 Bioanalyzer. Qualified libraries were sequenced using an Illumina NovaSeq 6000 platform (Illumina, San Diego, CA, USA). Image data from the high-throughput sequencer were converted into sequence data (reads) using CASAVA [14] base recognition. Reads containing ambiguous bases and low-quality reads were removed to retain only clean reads. The clean reads of mRNA sequencing and small RNA sequencing were mapped to annotation files using Hisat2 (v2.0.5) and Bowtie [15] (0.12.9), respectively.

### 2.3. DEG Analysis and Novel Transcript Prediction

To analyze the DEGs between the two groups, the R software package DESeq2 [8] (1.20.0) was used to determine mRNA expression levels. The resulting *p*-values were adjusted (padj) using the Benjamin-Hochberg method for controlling the false discovery rate, and padj ≤ 0.05 and log2FC (fold change) ≥ 1 or ≤ −1 were set as the threshold for significantly differential expression. Novel transcripts were assembled using StringTie [15] (v1.3.3b).

### 2.4. DEM Analyses and Target Gene Prediction

Differential expression analysis of the two groups was performed using the DESeq R package (1.24.0), and the *p*-value was adjusted using the Benjamin-Hochberg method. A corrected padj ≤ 0.05 was set as the threshold for differentially expressed genes. Novel miRNAs were predicted using miREvo [16] (v1.1) and miRdeep [3] software, because the characteristics of the hairpin structure of miRNA precursors can be used to predict novel miRNAs. TargetScan [3] (http://www.targetscan.org; accessed on 29 December 2021), miRDB [11] (http://www.mirdb.org/; accessed on 29 December 2021), and miRwalk [11] (http://mirwalk.uni-hd.de/; accessed on 29 December 2021) databases were used to predict the targets of the DEMs, and the targets that were common in the above software were determined as final targets for further analysis.

### 2.5. Gene Ontology and Kyoto Encyclopedia of Genes and Genomes Analyses

Gene Ontology (GO) and Kyoto Encyclopedia of Genes and Genomes (KEGG) enrichment analyses of DEG and DEM targets were implemented using the R package ClusterProfiler [8,9] (3.8.1). Differences were considered statistically significant at padj ≤ 0.05.

### 2.6. LncFAM200B-miRNA–mRNA Interaction Network Construction

Communal genes that appeared in the DEG and DEM target analyses were selected to construct the *lncFAM200B*-miRNA-mRNA interaction network. The up-regulated and down-regulated genes were analyzed separately, and Cytoscape software (v3.6.1) was used to visualize the interaction of the ceRNA network.

### 2.7. Quantitative Recerce-Transcription PCR

The total RNA was extracted using TRIzol method, and then cDNA was produced with the PrimeScript RT Reagent Kit (RR047A, Takara Bio, Shiga, Japan). Stem-loop RT primers were designed for miRNA cDNA synthesis, according to a previous study [17]. Primer information is listed in Appendix A. Six down-regulated miRNAs and ten up-regulated expression targets in the ceRNA network and other related genes were determined using the SYBR Premix Ex Taq kit (RR820A, Takara Bio). *GAPDH* and *U6* served as internal references to normalize gene expression levels and miRNA levels, respectively, via the 2^−∆∆Ct^ method.

### 2.8. Cell Transfection, Proliferation Detection, and Flow Cytometry

MiR-6529a mimics and inhibitors were synthesized according to the bta-miR-6529a mature sequence (Access ID: MIMAT0025565). When preadipocytes reached a confluence of approximately 70–80%, mimics or inhibitors (using the respective NC as control) were transfected into cells according to a previously described transfection procedure [6]. After 48 h of culturing, the cells were harvested for detection. Flow cytometry was performed on a Sysmex Cube 8.0 (Sysmex, Kobe, Japan) platform. Briefly, the cells were washed with pre-cooled PBS solution, centrifuged (1500 rpm × min^−1^ for 5 min), and fixed over night with 70% absolute ethanol at 4 ℃. Then, 50 ng × mL^−1^ PI reagent (Solarbio, Beijing, China) was added for incubation for 30 min at room temperature after washing with PBS solution. During CCK-8 detection, 10 µL of CCK-8 reagent (Tiangen, Beijing, China) was added to each well, followed by incubation for 1h at 37 °C; absorbance was measured at 510 nm. Sterile pipette tips were used to scratch along the central axis of the orifice plate at a uniform speed, and scratch width was recorded on different incubation days.

### 2.9. Cell Differentiation Induction and Oil Red O Staining

To test the effect of miR-6529a on lipid position during preadipocyte differentiation, preadipocytes were induced to differentiate for six days before miR-6529a overexpression or suppression. Oil Red reagent (Solarbio) was used to measure the amount of cytoplasmic lipid deposition on day 8. The cells were then washed with PBS and were fixed with 4% paraformaldehyde for 1 h. The fixed cells were washed with sterile ddH_2_O, stained with Oil Red staining solution at room temperature for 30 min, and photographed after being washing five times using sterile ddH_2_O.

### 2.10. Statistical Analysis

One-way ANOVA was performed using SPSS (version 25.0; IBM Corp, Armonk, NY, USA), and significance was determined using Duncan’s test. Significance was reported at *p* ≤ 0.05.

## 3. Results

### 3.1. DEGs Analysis after lncFAM200B Overexpression in Preadipocytes

On average, 43,605,862 and 47,328,137 raw reads were obtained from the Ad_lnc200B and Ad_G libraries, respectively, and 42,283,436 and 45,860,422 clean reads, on average, remained after filtering (Appendix A). The clean reads were uniquely aligned to the yak genome (*BosGru* 3.0), and the average mapping rations were 85.94% and 85.79%, respectively.

After screening and comparing the alignment results between *lncFAM200B* overexpression and control groups, 118 DEGs were identified (Figure 1A). The 76 significantly up-regulated genes included interferon-responsive genes (*IFI44L*, *IFI6*, *OAS1*, *OAS2*), IFIT family members (*IFIT1*, *IFIT2*, *IFIT3*, *IFIT5*), ubiquitin-related genes (*ISG15*, *PML*, *TRIM21*, *UBA7*, *TRIM25*), and adipogenesis regulator *NOTCH3*. Meanwhile, 42 significantly down-regulated genes were related to cell cycle and proliferation, including kinin family members (*KIF20A*, *KIF11*, *KIF2C*, *KIF23*), centromere protein members (*CENPF*, *CENPN*), cell cycle regulator cyclin A2 (*CCNA2*), cell division cycle-associated protein (*CDC20*, *CDCA2*), and DNA topoisomerase II (*TOP2A*). Furthermore, 17 novel genes and transcripts were identified. (Figure 1B, Appendix A).

KEGG pathway and GO enrichment analyses were performed to assess signaling pathways or biological functions regulated by the DEGs. KEGG pathway analysis showed that the up-regulated DEGs were significantly enriched in the cytosolic DNA-sensing and Notch signaling pathways (Figure 1C), and down-regulated DEGs were enriched in the Oocyte meiosis and Cell cycle pathways (Figure 1D). GO enrichment analyses revealed that these DEGs were involved in microtubule-based processes and movements, GTPase activity, hydrolase activity, transferase activity, and NAD+ ADP-ribosyltransferase activity (Figure 1E). These results imply that *lncFAM200B* may affect the synthesis and activity of proteases, activate the Notch signaling pathway, silence the cell cycle pathway, and regulate cell process.

### 3.2. Differentially Expressed miRNAs after lncFAM200B Overexpression in Preadipocytes

To analyze the miRNA profiles during *lncFAM200B* effects on preadipocyte differentiation, the small RNA sequencing was performed. The results showed that most reads were approximately 22–23 nt, and 23 nt length was most frequent (Appendix A). An average of 483 and 492 sRNA sequences in the Ad_ G and Ad_lnc200B groups, respectively, were mapped to mature miRNAs in the miRBase (www.mirbase.org, accessed on 26 June 2022) database, and 78 and 85 novel miRNAs were predicted in the two groups, respectively (Appendix A). A total of 610 miRNAs and 138 novel miRNAs were identified, and 619 common miRNAs were screened (Figure 2A); 25 differentially expressed miRNAs were identified, which included 13 up-regulated and 12 down-regulated miRNAs (Figure 2B, Appendix A).

Because miRNAs generally play a negative role in regulating mRNA in organisms, we predicted the targets of these DEMs, resulting in 6614 predicted potential targets. KEGG and GO analyses were performed on the targets to analyze the potential biological functions of the DEMs (Appendix A). The targets were enriched in the MAPK, mTOR, PI3K-Akt, and FoxO signaling pathways, insulin resistance, and other fat metabolism-related pathways. We found that targets of 10 down-regulated miRNAs and five up-regulated miRNAs participated in the regulation of these pathways (Figure 2C). Moreover, GO annotation showed that most targets were involved in components of the nucleus and cytoplasm and functions of metal ion and ATP binding, which were regulated by nine down-regulated miRNAs and six up-regulated miRNAs (Figure 2D). These results suggest that *lncFAM200B* may serve as an endogenous ceRNA sponge by interfering with those DEMs, and then regulate preadipocyte differentiation through the above pathways.

### 3.3. LncFAM200B as a Core of the Regulatory Network Construction

To connect DEMs and DEGs, a regulatory network of *lncFAM200B* in preadipocyte differentiation was constructed by comparing 118 DEGs and 6614 potential target genes. We found 29 common genes in both data sets (Appendix A), which contained 17 up-regulated DEGs and 12 down-regulated DEGs in the mRNA sequencing results (Table 1), and these 29 DEGs were predicted to be regulated by 10 DEMs (Figure 3). The Notch signaling pathway members *HES4* and *DTX3L* are regulated by miR-6529b and *NOTCH3* is regulated by miR-148c; *FBXO33*, an ubiquitination modulator [18], is regulated by miR-380-3p, miR-152, miR-27a-5p, and miR-6529a, and miR-6529a regulates the E3 ubiquitin ligase *TRIM21*. Furthermore, it is worth noting that many other DEGs have been reported to be involved in cell proliferation or differentiation-related regulation, including *ADAR*, which is essential to differentiation and proliferation of cells [19,20]. *AGRN* can elevate the activity of the WNT pathway by increasing cell cycle-related gene expression to inhibit suppressed rectal cancer cell growth [21].

### 3.4. RT-qPCR Results

Six down-regulated DEMs and 10 targets were selected for assessing their expression levels through RT-qPCR. The results showed that the expression levels of bta-miR-152, bta-miR-6529a, bta-miR-6529b, bta-miR-148c, bta-miR-380-3p, and bta-miR-27a-5p were significantly decreased (Figure 4A, *p* < 0.05), and the targets were significantly increased after *lncFAM200B* overexpression in preadipocytes (Figure 4B, *p* < 0.05). These results were consistent with the sequencing results, suggesting high reliability and accuracy of the data obtained in this study.

### 3.5. Effects of miR-6529a on Preadipocytes Differentiation and Proliferation

Based on the mRNA sequencing results, we speculated whether an association exists between the proteins encoded by DEGs; thus, a protein-protein interaction (PPI) network was analyzed via STRING database (https://cn.string-db.org/; accessed on 9 April 2022). The results (Figure 5) showed that the interaction was separated into two main parts: one of the interactions was associated with the ubiquitin-proteasome system, the members of which were encoded by up-regulated genes. The other interaction was associated with cell cycle-related proteins, which were encoded by down-regulated genes. These results suggested that *lncFAM200B* regulates molecular regulation by controlling ubiquitination and cell cycle-related processes. In addition, the Notch pathway, which was enriched in DEG functional annotations, also produced a connection with the ubiquitin-mediated proteolysis pathway (Figure 6). Therefore, we chose bta-miR-6529a, an upstream regulator of the E3 ubiquitin ligase *TRIM21*, to further examine its effects on preadipocyte differentiation and proliferation (the binding site is shown in Appendix A).

We successfully achieved up-regulation of miR-6529a expression in cells by transfection of miR-6529a mimics in preadipocytes (Figure 7A). The expression of its target, *TRIM21*, was significantly suppressed, whereas the expression level of *lncFAM200B* was not affected (Figure 7B). The scratch test and CCK-8 assay results showed that the cell proliferation rate was significantly decreased after overexpression of miR-6529a (Figure 7C,D). Moreover, flow cytometry showed that overexpression of miR-6529a significantly increased the number of G0/G1 phase cells (*p* < 0.05) and reduced the number of cells in the S phase (Figure 7E, *p* < 0.01). Furthermore, proliferation and adipocyte differentiation marker genes were significantly decreased (Figure 7F,H). In addition, the Oil Red O staining showed that cell lipid deposition was significantly reduced (Figure 7G). These results suggest that miR-6529a overexpression in yak preadipocytes inhibits the cell proliferation and differentiation.

To verify the inhibitory effects of miR-6529a on proliferation and differentiation of preadipocytes, we further inhibited the expression of miR-6529a in preadipocytes (Figure 8A). Transfection with the miR-6529a inhibitors significantly up-regulated *TRIM21* expression but had no effect on *lncFAM200B* expression (Figure 8B). Compared to the control group, the scratch recovery rate and the activity of CCK-8 increased significantly after miR-6529a down-regulation (Figure 8C,D). Flow cytometry showed that the inhibition of miR-6529a mainly increased S phase cells (Figure 8E, *p* < 0.05). In addition, expression levels of proliferation (*CCNA2* and *CCND1*) and differentiation (*PPARG* and *C/EBPα*) marker genes were significantly increased (Figure 8F,H), and the Oil Red O staining also indicated that lipid deposition was reduced after downregulation of miR-6529a (Figure 8G). These results indicate that miR-6529a downregulation can promote preadipocyte proliferation, differentiation, and lipid deposition.

## 4. Discussion

In the present study, we initially identified 118 mRNAs that were significantly differentially expressed following *lncFAM200B* overexpression. To further confirm the regulatory factors of *lncFAM200B* during lipid deposition in yak preadipocytes, we annotated the functions of the DEGs. The KEGG results showed that the Notch signaling activity of cells was significantly improved after *lncFAM200B* overexpression (Figure 1C and Figure 8), which included *NOTCH3*, *HES4*, and *DTX3L*. The Notch signaling pathway plays an important role in adipocyte differentiation, but its regulatory role remains controversial [22]. A previous study found that the Notch signaling pathway is mainly involved in early embryonic development and related metabolic processes in cells, and it causes fat deposition [23]. In human adipose stem cells, *NOTCH3* knockout significantly down-regulates cell fat deposition and significantly reduces the expression of marker genes related to adipocyte differentiation [24]. Furthermore, as classical target genes of Notch signaling, the *Hairy Enhancer of Split* (*HES*) family plays an important role in metabolism [23]. *HES4* can promote the induction of human T-lineage differentiation when Notch signaling occurs at higher levels [25], and it is involved in regulating the differentiation of photoreceptors in zebrafish [26]. In the current study, overexpression of *lncFAM200B* significantly up-regulated the expression levels of *NOTCH3* and *HES4* during preadipocyte differentiation, and *NOTCH3* and *HES4* were regulated by miR-148c and miR-6529b, respectively. Therefore, *lncFAM200B* may promote yak preadipocyte differentiation and lipid deposition by sponging miR-6529b and miR-148c, which promotes *NOTCH3* and *HES4* expression and enhances Notch signaling activity. Consequently, our next step was to verify the regulatory relationship between *lncFAM200B* and the Notch signaling pathway and to identify the downstream targets of Notch signaling.

In previous studies, cell proliferation and differentiation typically exhibited mutual inhibition of biological processes [27,28]. *LncFAM200B* not only promoted preadipocyte differentiation but also inhibited cell proliferation in our previous study. However, bta-miR-6529a, which has an obvious binding site with *lncFAM200B*, had a negative regulatory role in either proliferation or differentiation in yak preadipocytes in the present study. This may be related to the mechanism of *TRIM21*, a target of miR-6529a, which promotes glioma cell growth and suppresses cell senescence through the p53-p21 pathway [29]; however, the specific mechanism requires further verification. TRIM21 can control the process of ubiquitination and degradation of FASN to regulate lipid metabolism [30], negatively regulate the crosstalk between the PI3K/AKT pathway and PPP metabolism [31], and affect cell apoptosis and proliferation [32]. Cell proliferation is influenced by a variety of cell cycle regulators [33], and our sequencing data showed that *lncFAM200B* overexpression in preadipocytes decreased the expression levels of cell cycle-related genes (Figure 1B), including *CCNA2*, *BUB1*, *KIF20A*, and *TOP2A*. CCNA2, a member of the cyclin family, is known to exhibit periodic protein expression throughout the cell cycle [34]. CCNA2 regulates the cell cycle by promoting transition through G1/S and G2/M phases, and knockdown of *CCNA2* inhibits trophoblast proliferation [35]. CDC20 is a regulatory factor of cell cycle checkpoints, which are initiated in the late cell cycle and exit in late mitosis. TOP2A accumulates in the nucleus during the G2 phase of the cell cycle; all of these play active regulatory roles in the development of the cell cycle and promote cell proliferation [36]. Budding uninhibited by benzimidazoles 1 (BUB1) is a mitotic checkpoint serine/threonine kinase, which promotes the proliferation of lung cancer cells [37]. In addition, *CENPN*, *CENPF*, *KIF2C*, *KIF11*, *CDCA2*, and *KIF23* were included (Appendix A). Overall, the above results suggest that *lncFAM200B* may arrest cell cycle progression by inhibiting the expression of these cell cycle-related genes and ultimately inhibit proliferation of preadipocytes.

Our findings showed that *lncFAM200B* significantly enhanced the Notch signaling pathway and decreased the expression of cell cycle-related genes. Strikingly, deltex-3-like (*DTX3L*) was up-regulated in the Notch signal pathway (Figure 6); it is an E3 ubiquitin ligase and plays a key role in the cell-cycle-related processes and cell-cycle regulation [38]. DTX3L belongs to the DTX protein family and is closely associated with cell signal transduction, growth, differentiation, and apoptosis [39]. DTX3L forms a complex with PARP14 and together with PARP14 and PARP9 mediates proliferation [40]. In particular, *PARP14* and *PARP9* were significantly up-regulated after *lncFAM200B* overexpression; therefore, we propose an interaction between the E3 ubiquitin ligase DTX3L and poly (ADP-ribose) polymerases (PARPs) during the differentiation of yak preadipocytes. PARP9 undergoes heterodimerization with DTX3L, and the DTX3L/PARP9 heterodimer mediates NAD+dependent mono-ADP-ribosylation of ubiquitin [41]. This molecular function was also reflected in our GO analysis (Figure 1E): NAD+ ADP-ribosyltransferase activity was significantly enhanced, which is a key factor in the maintenance of PARP activity [42].

The PARP family of 17 proteins is involved in a wide range of biological processes [43,44], and many of the 17 PARPs are involved in lipid metabolism, including PARP2 and PARP7 [43]. In our RNA-Seq data, four PARP genes were significantly up-regulated (*PARP9*, *PARP10*, *PARP12*, and *PARP14*; Appendix A). Intriguingly, ADP-ribosyltransferases are closely associated with the ubiquitin-proteasome degradation system. A previous study showed that the C-terminus of ubiquitin is the site of ADP-ribosylation, and ADP-ribose can also bind to the C-terminal sequence of ubiquitin-related protein ISG15 [41]. ISG15 is an ubiquitin-like proteins, and its E1 ubiquitin ligase is UBA7 [45]. Meanwhile, both showed high mRNA expression levels in the *lncFAM200B* overexpression group (Figure 8, Appendix A), and there was an interaction relationship at the proteins (Figure 5), which implies that *lncFAM200B* affects cell progression by regulating the ubiquitination process in preadipocytes.

In the ubiquitin mediated proteolysis pathway (Figure 8), the anaphase-promoting complex/cyclosome (APC/C) plays an important role regarding the regulation of the cell cycle with CCNA2 as an ubiquitination substrate [42]. However, ubiquitination of substrates by APC/C requires activation or co-activation of CDC20 or CDH1 [46,47,48]. In addition to CCNA2 regulating cell proliferation by binding with CDC20 [49], CCNA2 is also regulated by CSN1 (signalosome subunit 1) in an ubiquitination-independent manner, which affects the proliferation and migration of hepatocellular carcinoma cells [50]. Surprisingly, the PPI network (Figure 5) showed that the proteins controlling cell cycle progression (mRNAs were down-regulated, according to sequencing results) can interact with ubiquitin-proteasome-related proteins (mRNAs were up-regulated), which implies that ubiquitination of cell cycle-related proteins (such as CCNA2 and CDC20) determines cell development, and ubiquitination plays an indispensable role in the regulation of yak preadipocyte proliferation and differentiation by *lncFAM200B*.

In summary, bta-miR-6529a, a target of *lncFAM200B*, plays a negative role in regulating the proliferation and differentiation of yak preadipocytes, and the ubiquitination regulation may play a role in preadipocyte differentiation and proliferation. In addition, overexpression of *lncFAM200B* significantly enhanced the activity of the Notch signaling pathway during yak preadipocyte differentiation. Mechanistically, the *lncFAM200B*-miR-6529b-*HES4*/*DTX3L* and *lncFAM200B*-miR-148c-*NOTCH3* regulatory axis plays an important role during preadipocyte differentiation. Following *lncFAM200B* overexpression, the expression of cell cycle-related genes (*CCNA2*, *BUB1*, *CDC20*, *TOP2A* and *KIF20A*) was reduced, which are key regulators *lncFAM200B* inhibiting the preadipocyte proliferation.

## Figures and Tables

**Figure 1 cells-11-02366-f001:**
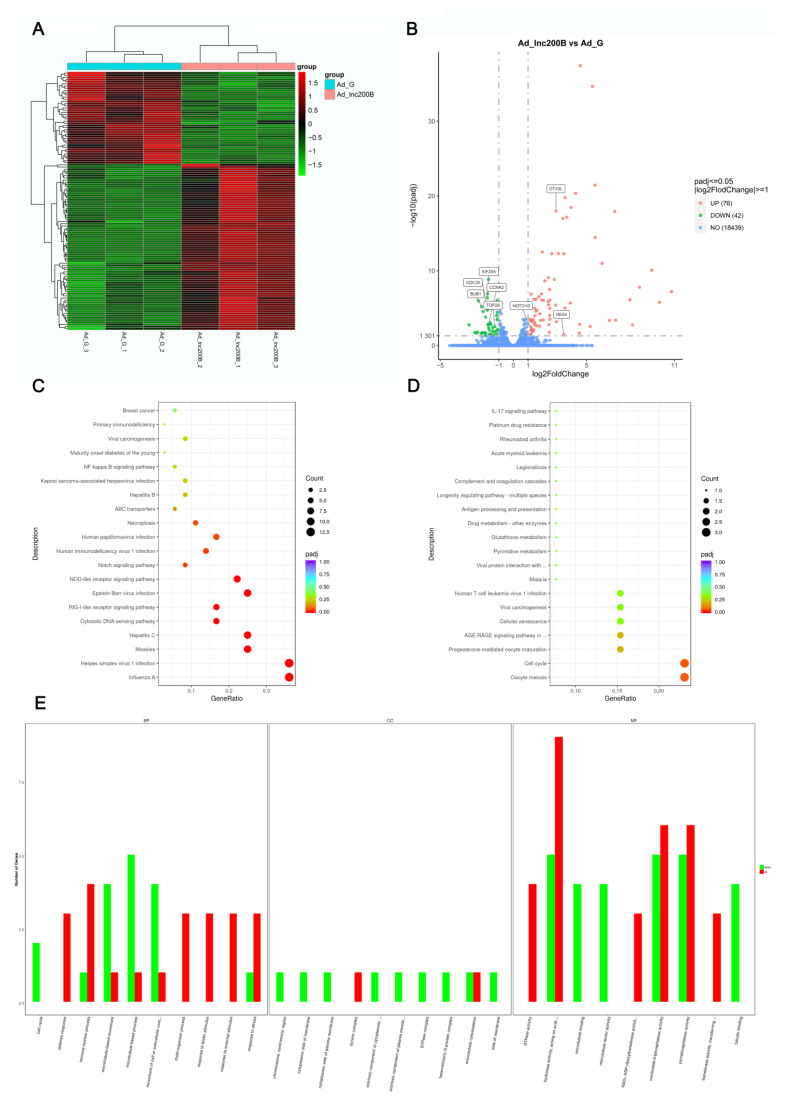
DEGs analysis and its functions annotation after *lncFAM200B* overexpression. (**A**) Clustering map of DEGs. (**B**) Volcanic map of DEGs. The up-regulation genes of the Notch signaling pathway members and down-regulation of cell cycle related genes are mapped. The top 20 enriched pathways of up-regulated and down-regulated DEGs are presented in (**C**,**D**), respectively. (**E**) GO enrichment analysis of DEGs. The GO terms of up-regulated and down-regulated DEGs was differentiated by red and green color, respectively.

**Figure 2 cells-11-02366-f002:**
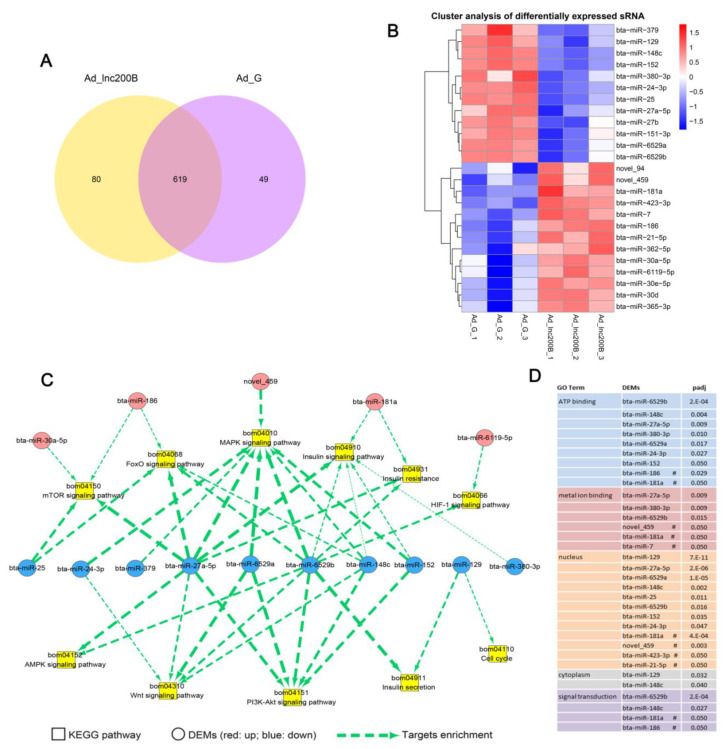
DEMs analysis after *lncFAM200B* overexpression. (**A**) The Venn diagrams of the DEMs between Ad_lnc200B and Ad_G. (**B**) Clustering map of DEMs. Statistics of adipogenesis-related pathway and main GO terms in DEMs targets. (**C**) Analysis of DEMs in adipogenesis-related KEGG pathway. The thickness of the line is negatively correlated with the padj. (**D**) Main GO term in the DEMs target. # marks the up-regulated DEMs, the others are down-regulated DEMs.

**Figure 3 cells-11-02366-f003:**
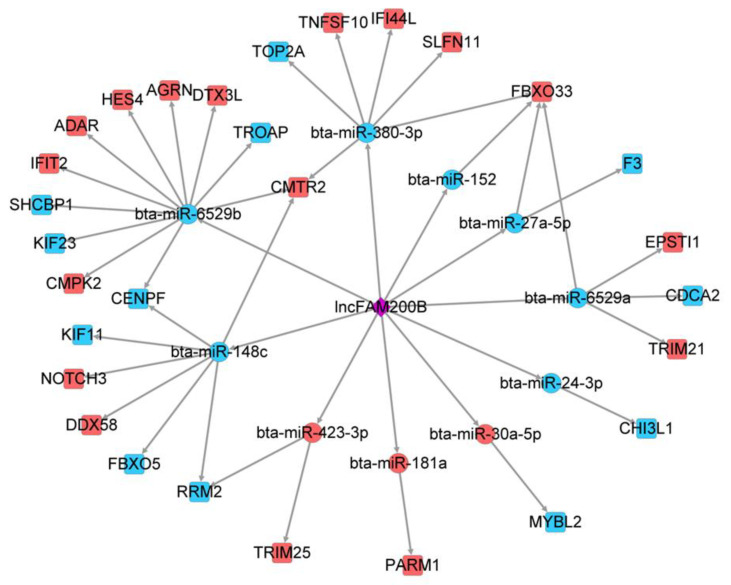
The ceRNAs network of *lncFAM200B* during the yak preadipocyte differentiation. The red color indicates up-regulated DEMs and DEGs, and the blue color indicates the down-regulated DEMs and DEGs after *lncFAM200B* overexpression.

**Figure 4 cells-11-02366-f004:**
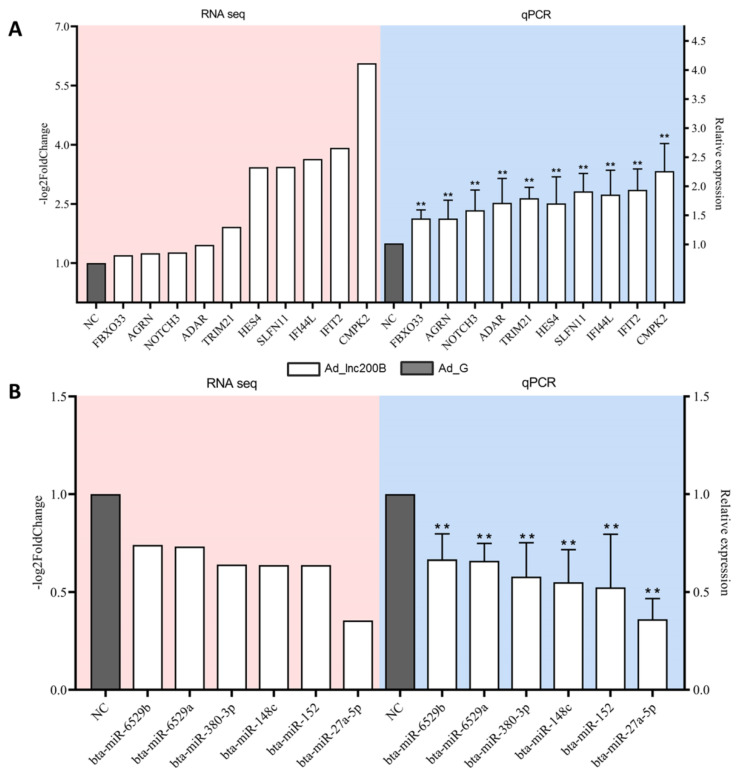
The validation of the mRNA and miRNA expression after *lncFAM200B* overexpression. (**A**) The mRNA expression levels of DEGs. (**B**) The miRNA expression levels of DEMs. Results are presented as mean ± standard deviation (SD) from three independent experiments. ** *p* < 0.01.

**Figure 5 cells-11-02366-f005:**
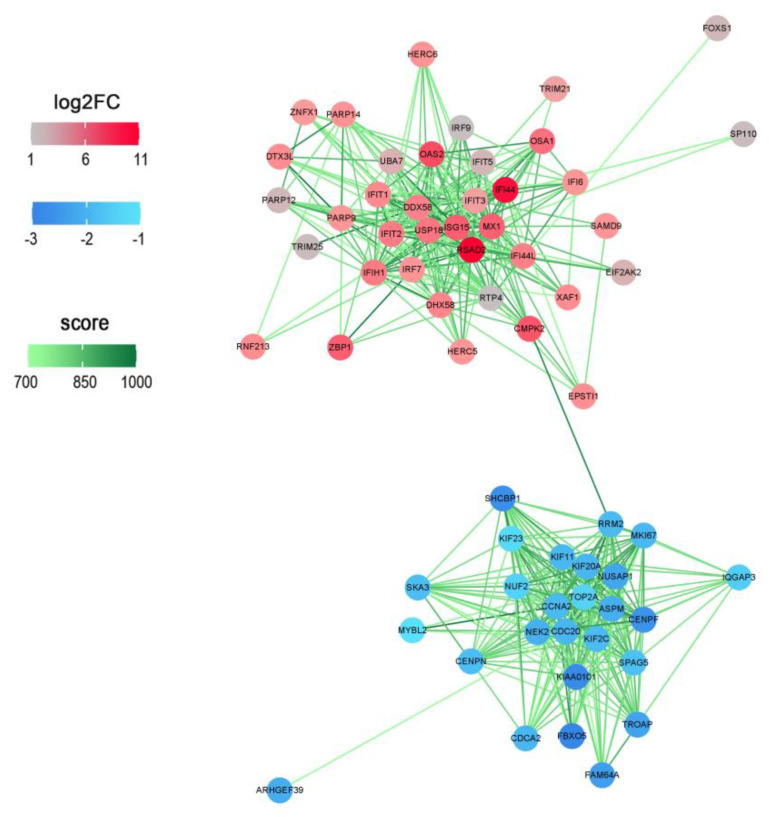
PPI network of DEGs coding proteins. The green scale indicates the STRING database prediction score; the red scale and the blue scale represent the up-regulated and down-regulated DEGs in the mRNA-Seq results, respectively.

**Figure 6 cells-11-02366-f006:**
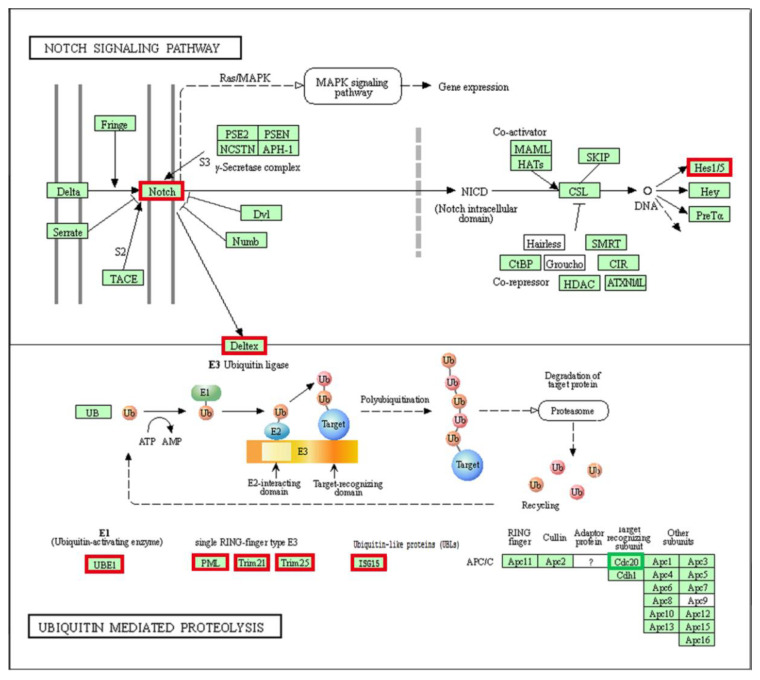
The DEGs in Notch signal pathway and ubiquitin-mediated proteolysis pathway. The red box indicates the up-regulated genes in the pathway after *lncFAM200B* overexpression, and the green indicates down-regulation.

**Figure 7 cells-11-02366-f007:**
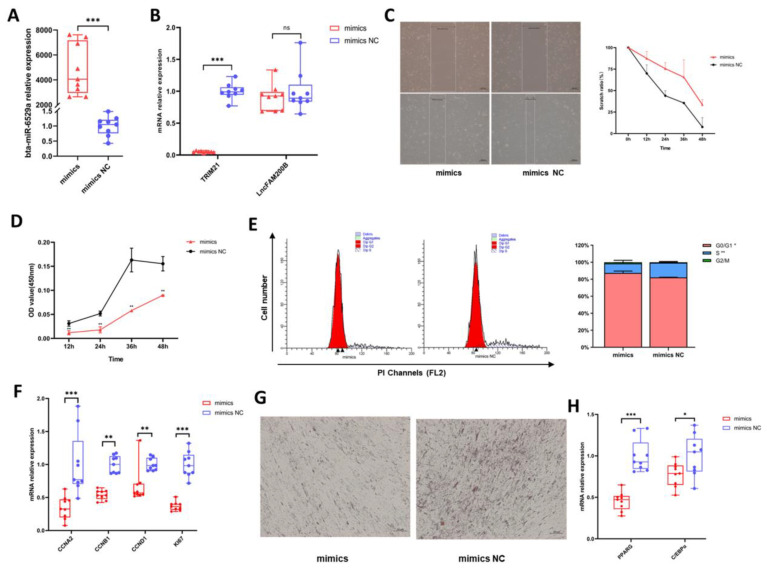
Effects of up-regulation of miR-6529a expression on proliferation and differentiation of preadipocytes. (**A**) Overexpression of miR-6529a. (**B**) The mRNA expression levels of *TRIM21* and *lncFAM200B* after miR-6529a overexpression. (**C**) Scratch test. Only 0 h (top) and 36 h (bottom) results are posted here. The Scratch ratio was obtained by dividing the current scratch width by the initial width. (**D**) CCK-8 results. (**E**) Flow cytometry after miR-6529a overexpression. (**F**,**H**) are the expression of proliferation marker and differentiation marker after overexpression of miR-6529a, respectively. (**G**) Oil red O staining results. Results are presented as mean ± SD, * *p* < 0.05. ** *p* < 0.01. *** *p* < 0.001.

**Figure 8 cells-11-02366-f008:**
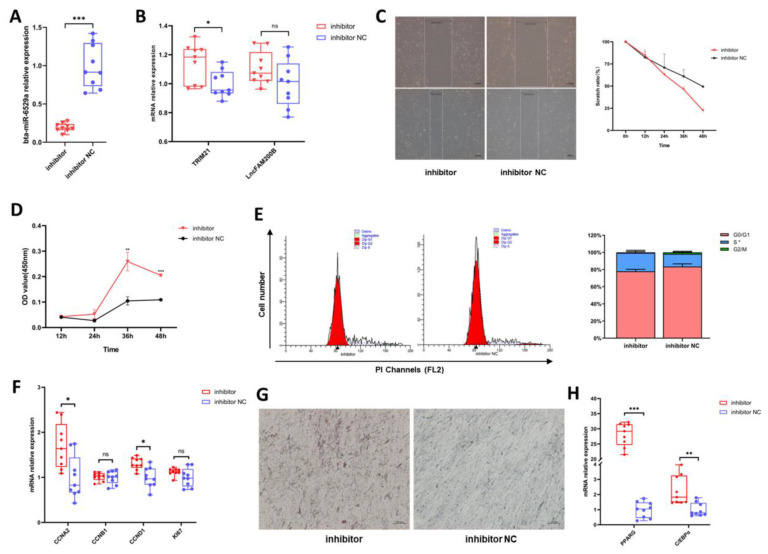
Effects of suppression of miR-6529a expression on the proliferation and differentiation of preadipocytes. (**A**) Inhibition of miR-6529a. (**B**) The mRNA expression levels of *TRIM21* and *lncFAM200B* after miR-6529a inhibition. (**C**) Scratch test. The top figures are results at 0h and the bottom figures are results at 36h. (**D**) CCK-8 results. (**E**) Flow cytometry after miR-6529a inhibition. (**F**,**H**) are the expression of proliferation marker and differentiation marker after inhibition of miR-6529a, respectively. (**G**) Oil red O staining results. Results are presented as mean ± SD, * *p* < 0.05. ** *p* < 0.01. *** *p* < 0.001.

**Table 1 cells-11-02366-t001:** The differential expression level of 29 common DEGs.

Gene ID	Gene Name	log2FC	padj
ENSBGRG00000001303	*DDX58*	3.527	1.64 × 10^−20^
ENSBGRG00000017758	*IFIT2*	3.927	3.38 × 10^−19^
ENSBGRG00000007487	*DTX3L*	2.906	1.05 × 10^−18^
ENSBGRG00000002173	*IFI44L*	3.643	7.02 × 10^−18^
ENSBGRG00000022760	*SLFN11*	3.444	5.33 × 10^−13^
ENSBGRG00000015652	*CMPK2*	6.067	1.06 × 10^−11^
ENSBGRG00000010858	*RRM2*	−1.706	1.46 × 10^−9^
ENSBGRG00000004611	*EPSTI1*	2.798	1.47 × 10^−9^
ENSBGRG00000012893	*TRIM21*	1.922	1.23 × 10^−7^
ENSBGRG00000018577	*CMTR2*	1.552	6.83 × 10^−7^
ENSBGRG00000026316	*CHI3L1*	−1.107	9.32 × 10^−7^
ENSBGRG00000019358	*CENPF*	−2.395	1.05 × 10^−6^
ENSBGRG00000000221	*F3*	−1.110	3.73 × 10^−5^
ENSBGRG00000015397	*AGRN*	1.251	0.0004
ENSBGRG00000003840	*TNFSF10*	6.983	0.0004
ENSBGRG00000017269	*TRIM25*	1.286	0.0007
ENSBGRG00000008643	*ADAR*	1.461	0.0016
ENSBGRG00000020999	*KIF11*	−1.743	0.0016
ENSBGRG00000019090	*PARM1*	1.795	0.0026
ENSBGRG00000017633	*TOP2A*	−1.268	0.0028
ENSBGRG00000007011	*FBXO33*	1.202	0.0034
ENSBGRG00000016906	*CDCA2*	−1.407	0.0043
ENSBGRG00000012371	*MYBL2*	−1.010	0.0091
ENSBGRG00000020219	*KIF23*	−1.146	0.0110
ENSBGRG00000018066	*NOTCH3*	1.277	0.0142
ENSBGRG00000020652	*SHCBP1*	−2.452	0.0173
ENSBGRG00000004593	*FBXO5*	−2.635	0.0187
ENSBGRG00000001616	*TROAP*	−2.112	0.0196
ENSBGRG00000016269	*HES4*	3.434	0.0332

## Data Availability

The data presented in this study are openly available in the NCBI BioProject database (https://submit.ncbi.nlm.nih.gov/subs/bioproject/, accessed on 26 June 2022). The accession number for mRNA-Seq and miRNA-Seq is PRJNA831610 and PRJNA831897, respectively.

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
