# Peer review of "Molecular Regulation of Yak Preadipocyte Differentiation and Proliferation by LncFAM200B and ceRNA Regulatory Network Analysis"

_cells, 2022, doi:10.3390/cells11152366_

Round 1

Reviewer 1 Report

In the presented work, the authors reported a regulatory network of lncFAM200B during yak preadipocytes differentiation. It is clear that a lot of effort has gone into producing this manuscript.

Basically, I am interested in your paper’s aim. However, I have doubts about choosing a PrimeScript RT Reagent Kit (RR047A, Takara Bio) for miRNA validation. I have not found on the manufacturer’s website that this kit is also dedicated to performing a reverse transcription of miRNA. Using the general kit for cDNA synthesis, you can synthesized the cDNA of mature miRNAs and their precursors, which later could distorted the analysis result. Please confirm that the PrimeScript RT Reagent Kit can be used for miRNA analysis. Otherwise, please re-validate miRNA expression using dedicated reagents.

The authors would also benefit from English language advice in order to rewrite the entire manuscript; there are far too many grammatical and other related issues for a reviewer to correct.

Section 2.2 - The total RNA library preparations pipeline is minimally and incompletely described. Please provide more information on the procedure for preparing libraries for NGS sequencing (including the name and manufacturer of the used reagents).

Section 3.2 - I do not fully understand where the authors obtained the 619 common miRNAs. From the description in the text, it appears that group Ad_G contained 483 mature miRNA and 78 novel miRNAs, and group Ad_lnc200B contained 492 mature miRNA and 85 novel miRNA. In turn, figure 3 shows that 668 miRNAs were identified in the Ad_G group and 699 miRNAs in the Ad_lnc200B group. Please explain why these differences in numbers?

Minor issues:

-lines 86, 355 - no spaces;

- line 141 - needs a reference

Author Response

In the presented work, the authors reported a regulatory network of lncFAM200B during yak preadipocytes differentiation. It is clear that a lot of effort has gone into producing this manuscript.

Response: We thank the reviewer for his/her positive and helpful comments on our work.

Point 1:Confirm that the PrimeScript RT Reagent Kit (RR047A, Takara Bio) can be used for miRNA cDNA synthesis and analysis

Response 1: We designed Stem-loop primers instead of the Primer mix in this kit to performed miRNA cDNA synthesis. Then quantitative analysis was performed with universal reverse and specific forward primers. We have added the reference about this method.

Point 2: “…there are far too many grammatical and other related issues for a reviewer to correct”

Response 2: We have corrected grammar errors and improved the manuscript.

Point 3: “Section 2.2 -… provide more information on the procedure for preparing libraries for NGS sequencing…”

Response 3: We have added this information in the revised manuscript.

Point 4: “Section 3.2 – Please explain why the number of miRNAs previously obtained in Ad_G and Ad_lnc200B is different to the final?”

Response 4: We apologize that the misunderstanding in the description in the manuscript. In fact, there are three sample replicates in each treatment group. 483/78 and 492/85 are the average number of each group (the number of each sample is shown in table S1). Because there may experimental error between different samples in the same group, therefore, we integrated these miRNAs together and obtained a total of 610 different match miRNAs and 138 novel miRNAs. Subsequently, 619 common miRNAs between Ad_G and Ad_lnc200B group has been screened out. We have further detail described this part in the revised manuscript.

Point 5:“Minor issues:

-lines 86, 355 - no spaces;

-line 141 - needs a reference”

Response 5: I apologize for our sloppiness, we have reviewed and modified.

Reviewer 2 Report

In the manuscript, the authors indicated that lncFAM200B may arrest the cell cycle progress via inhibiting the expression of cell cycle  genes, and therefor inhibit the  proliferation of preadipocytes.

Figure 2C, the x Axis labeling needs to be more clear.

Was the enrichment of d NOTCH3, HES4, and DTX3L observed at the protein level?

Were any protein western blots performed? I would like to see the effect of up regulation/inhibition of miR-6529a on the protein of TRIM21.

Did you evaluate the direct target of miR-6529a?

How many replicates were used for the RNA Seq analysis?

Is the expression of LncFAM200B tissue specific? Has the evaluation of the lncRNA been evaluated in other tissues?

Some of the initial RNA seq analysis figures can be combined. 

Author Response

“In the manuscript, the authors indicated that lncFAM200B may arrest the cell cycle progress via inhibiting the expression of cell cycle genes, and therefor inhibit the proliferation of preadipocytes.

Response: We thank the reviewer for the helpful suggestion. I will address the reviewer’s concerns point by point below.

Point 1: “Figure 2C, the x Axis labeling needs to be more clear”

Response 1: We have corrected the figure Axis in the revised manuscript.

Point 2: “Was the enrichment of d NOTCH3, HES4, and DTX3L observed at the protein level?”

Response 2: I fully understand the concerns of the reviewer and apologize for not performed this part of the experiment. We tried to search the antibodies for this experiment, but unfortunately, we didn't find suitable antibodies that can be applied to yaks.

Point 3: “Were any protein western blots performed? I would like to see the effect of up regulation/inhibition of miR-6529a on the protein of TRIM21”

Response 3: I agree that we should verify the expression of TRIM21 at the protein level. However, the commercialized TRIM21 antibody which can interact with yak proteins has also not been found. Our next work will prepare the TRIM21 antibody and further to analyze its function from the protein level.

Point 4: “Did you evaluate the direct target of miR-6529a?”

Response 4: This is our ongoing work. In the future, we will further identify the direct target of miR-6529a and further explore the function of the targeted gene to construct the ceRNA network during the proliferation and differentiation of yak preadipocytes.

Point 5: “How many replicates were used for the RNA Seq analysis?”

Response 5: Three independent biological samples were used.

Point 6: “Is the expression of LncFAM200B tissue specific? Has the evaluation of the lncRNA been evaluated in other tissues?”

Response 6: In our previous study, we have carried out the expression profile of lncFAM200B between different ages and different tissues (Reference 12), and the results show that lncFAM200B has the highest expression in liver and lung, with some tissue specificity.

Point 7: “Some of the initial RNA seq analysis figures can be combined”

Response 7: As suggested, we have combined the result figures of mRNA-Seq and miRNA-Seq sequencing, respectively.

Round 2

Reviewer 1 Report

The authors have addressed all of my concerns. I recommend the presented manuscript for publication.

Reviewer 2 Report

The authors have made the relevant changes.